# Chronological Appearance of Endocrine and Metabolic Dysfunctions Induced by an Unhealthy Diet in Rats

**DOI:** 10.3390/medicina58010008

**Published:** 2021-12-21

**Authors:** María Cecilia Castro, Hernán Gonzalo Villagarcía, Carolina Lisi Román, Bárbara Maiztegui, Luis Emilio Flores, Guillermo Raúl Schinella, María Laura Massa, Flavio Francini

**Affiliations:** 1CENEXA, Centro de Endocrinología Experimental y Aplicada (UNLP-CONICET La Plata), La Plata 1900, Argentina or mccastro@cenexa.org (M.C.C.); hvillagarcia@med.unlp.edu.ar (H.G.V.); clroman@cenexa.org (C.L.R.); bmaiztegui@cenexa.org (B.M.); leflores@cenexa.org (L.E.F.); mlmassa@cenexa.org (M.L.M.); 2Facultad de Ciencias Médicas, Instituto de Ciencias de la Salud, Universidad Nacional de La Plata, UNAJ-CICPBA, La Plata 1900, Argentina; schinell@gmail.com

**Keywords:** fructose rich diet, prediabetes, liver and pancreatic islet oxidative stress

## Abstract

*Background and Objectives*: The work was aimed to determine the chronological sequence of events triggered by a fructose-rich diet (FRD) (10% *w*/*v* in the drinking water) in normal rats. *Material and Methods*: Serum parameters, liver and islet markers of metabolism, inflammation and oxidative stress were determined weekly for 21 days. *Results*: At the end of the first week, rats fed with a FRD showed an early increase in circulating triglycerides, fat liver deposit, and enzymatic activity of liver glucokinase and glucose-6-phosphate dehydrogenase (G6P-DH). After two weeks of such a diet, liver glucose-6-phosphatase (G6Pase) activity and liver oxidative stress markers were significantly increased. Liver sterol regulatory element-binding protein 1c (SREBP1c) mRNA also increased in the second week while their target genes fatty acid synthase (FAS) and glycerol-3-phosphate dehydrogenase (GPAT) enhanced their expression at the third week. Liver and pancreatic inflammation markers also enhanced their gene expression in the last week of treatment. Whereas both control and FRD rats remained normoglycemic throughout the entire period of treatment, blood insulin levels were significantly higher in FRD animals at the third week, thereby evidencing an insulin-resistant state (higher HOMA-IR, HOMA-B and HIS indexes). Pancreatic islets isolated from rats fed with a FRD for 3 weeks also increased glucose-induced insulin secretion (8.3 and 16.7 mM). *Conclusions:* FRD induces asynchronous changes involving early hypertriglyceridemia together with intrahepatic lipid deposit and metabolic disturbances from week one, followed by enhanced liver oxidative stress, liver and pancreas inflammation, pancreatic β-cell dysfunction, and peripheral insulin-resistance registered at the third week. Knowledge of time-course adaptation mechanisms involved in our rat model could be helpful in developing appropriate strategies to prevent the progression from prediabetes to Type 2 diabetes (T2D) triggered by unhealthy diets.

## 1. Introduction

Modern societies are characterized by a sedentary lifestyle accompanied by the consumption of unhealthy diets [1,2,3]. Since in recent decades, a heightened consumption of high fructose corn syrups (HFCS) has been evidenced [4,5], several authors have suggested a link between this trend and the epidemic of Type 2 diabetes (T2D) and obesity [3,5,6,7,8]. It has also been demonstrated that sweet beverages enriched in fructose increases the risk of obesity since its impact on satiety is lower in liquid presentations than in an isocaloric solid form [9,10,11]. This situation is even worse in children whose consumption of soft drinks is remarkably higher [7,12,13].

We previously showed that normal rats fed a fructose-rich diet (FRD) for three weeks developed changes in glucose and lipid metabolism together with endocrine dysfunction (hyperinsulinemia, hyperleptinemia, higher plasminogen activator inhibitor-1 and lower adiponectin levels) and an insulin resistant (IR) state [14,15,16,17,18,19,20]. All these changes suggest a complex multi-organ function compromise in animals fed a FRD: adipose tissue (evidenced by increased free fatty acids levels), liver (suggested by several alterations of carbohydrate metabolism and triglyceride levels), and endocrine pancreas (since hyperinsulinemia together with impaired glucose tolerance indicate β-cell functional compromise unable to cope with the enhanced demand for insulin due to IR).

However, the temporal course of the aforementioned changes remains unclear. Therefore, the aim of this study was to evaluate the time-course of events induced by oral fructose administration to normal rats. This knowledge, contributing to understanding the chronological appearance of adaptation mechanisms involved in fructose overload, could help to design effective strategies to prevent the progression from a prediabetic state towards T2D triggered by unhealthy diets.

## 2. Materials and Methods

### 2.1. Chemicals and Drugs

Reagents of the purest available grade and bovine serum albumin (BSA, fraction V) were from Sigma Chemical Co. (St. Louis, MO, USA). FastStart SYBR Green Master mix I were provided by Roche Diagnostics GmgH (Mannheim, Germany). Triglyceride color kit GPO/PAP AA was supplied by Wiener Laboratory (Buenos Aires, Argentina). DNase I and SuperScript III were provided by Gibco (Gibco-BRL, Waltham, MA, USA). Collagenase was obtained from Roche (Mannheim, Germany).

### 2.2. Animals

Normal male Wistar rats (180–200 g body weight) bred in fixed circadian conditions (12-h light-dark cycle) and constant room temperature (23 °C), were divided into two experimental groups: Control (C): fed with a standard solid diet and *ad libitum* tap water, and Fructose (F): fed with the same commercial diet and fructose in a final concentration of 10% *w*/*v* in tap water for one (C1 and F1), two (C2 and F2) and three weeks (C3 and F3), respectively. Each experimental group included 12 animals (three rats per group, replicated four times). Water intake was recorded daily whereas food intake and body weight were registered weekly. Commercial diet (solid food) consumed contained 62.8% of carbohydrates, lipids and proteins, while the remaining 37.2% was represented by fibers, vitamins, calcium and phosphorus. The carbohydrates: lipids: proteins ratio was 45:43:12, respectively. In the treated groups, addition of 10% fructose in the drinking water modified nutrient supply regarding control diet, by increasing carbohydrates content. Thus the carbohydrates: lipids:proteins ratio was modified in these animals. Daily calories consumed by each experimental group, were calculated considering the ingested grams of the different nutrients and the calories provided by each of them.

At each time-point (1, 2, and 3 weeks), blood samples were obtained from rats fasted for 4 h from the retroorbital plexus under halothane anesthesia, and glucose, triglyceride, and insulin levels were determined. Animals were euthanized by decapitation and whole pancreas and liver median lobe were carefully removed for assays.

### 2.3. Serum Measurements

Glycemic values were obtained from blood samples employing test strips (Accu-Chek Performa Nano System, Roche Diagnostics. Mannheim, Germany). Triglycerides and insulin levels were determined in serum by an enzymatic reaction kit (BioSystems S.A., Buenos Aires, Argentina) and radioimmunoassay [17], respectively. HOMA-IR (homeostasis model assessment for insulin resistance) and HOMA-β (homeostasis model assessment for β-cell function) were calculated as described by Mathews et al. [21] using the following formulas: HOMA-IR: serum insulin (µU/mL) × fasting blood glucose (mmol/L)/22.5) and HOMA-β: [insulin (μU/mL) × 20/glucose (mmol/L)] − 3.5. These indexes have been validated for measuring peripheral insulin sensitivity in rats [22,23]. Hepatic insulin sensitivity (HIS) was calculated according to Matsuda and DeFronzo [24] as follows: k/[fasting plasma insulin (µU/mL)] × fasting plasma glucose (mg/dL), where k: 22.5 × 18.

### 2.4. Protein Carbonyl Groups and Reduced Glutathione (GSH)

The carbonyl content of liver homogenate was determined by using a spectrophotometric method by derivatization with 2,4-dinitrophenylhydrazine (DNPH) as previously reported by Francini et al. [15]. The results were expressed as nmol of carbonyl residues/mg protein. GSH content was determined in the non-protein fraction of liver homogenates using the Ellman’s reagent and expressed as µmol GSH/mg of protein [15].

### 2.5. Liver Triglyceride Content

Hepatic triglycerides were extracted according to Schwartz and Wollins [25] and triglyceride levels were assayed with a commercial kit as described in Section 2.3.

### 2.6. Liver Glycogen Content

Liver medial lobe pieces were first incubated for 20 min at 100 °C in 1 mL 33% KOH, and further incubated for 24 h at 4 °C with the addition of 1.25 mL of 96° ethanol. Samples were later centrifuged for 20 min at 1200× *g* and the pellets resuspended in 1 mL distilled water with 3 mL of anthrone (100 mg diluted in 100 mL of 84% H_2_SO_4_). Finally, samples were incubated for 20 min at 100 °C and optical density was measured by photometry at 620 nm. Results were expressed as µmol glycogen/mg tissue [26].

### 2.7. Glucokinase (EC 2.7.1.2) Activity

Liver glucokinase activity was assayed as described elsewhere [27]. Briefly, liver pieces (300 mg) were homogenized in ice cold phosphate saline buffer and centrifuged at 600× *g* to discard the nuclear fraction. A soluble fraction was then centrifuged twice (at 8000× *g* and 100,000× *g* at 4 °C), and the resulting supernatant (cytosolic fraction was collected for phosphorylation measurements by recording the increase in absorbance at 340 nm in an enzyme-coupled photometric assay containing G6P-DH, ATP and NADP [27]. For glucokinase activity determination, the activity measured at 1 mM glucose (corresponding to hexokinase) was subtracted from that measured at 100 mM glucose (corresponding to hexokinase plus glucokinase). The current glucose concentrations employed (1 and 100 mM), were selected after trying out different glucose concentrations as previously described in Massa et al. [27]. Glucokinase activity was expressed as mU of enzyme per mg of protein.

### 2.8. Glucose-6-Phosphatase (G6Pase) Activity

For G6Pase activity, liver microsomes were isolated as described by Nordlie and Arion [28] in a homogenization medium containing 0.25 M sucrose/5 mM Tris-acetate/0.5 mM Na-EDTA, pH 7.4 (3 mL/g tissue). After being washed once with 0.25 M sucrose/5 mM Tris-acetate, pH 7.4, microsomes were centrifuged at 100,000× *g* and after that diluted to the desired final concentration with sucrose buffered solution.

Disrupted microsomes were obtained at 0 °C by adding 0.1 mL 0.75% (*w*/*v*) Triton X-100 to 0.9 mL of untreated microsomes (10 mg protein) and maintained on ice for 20 min. The reaction was stopped by adding to 200 µL of sample, 250 µL 10% TCA, 2 mL MoNH_4_/H_2_SO_4_ 1 N and 320 µL FeSO_4_/H_2_SO_4_ 0.15 N. Optical density was determined by photometry (660 nm). The results were expressed as “latency” [100 × (activity in disrupted microsomes—activity measured in untreated microsomes)/activity measured in disrupted microsomes] [29].

### 2.9. Glucose-6-Phosphate Dehydrogenase (G6P-DH) Activity

Hepatic pieces of the medial lobe (1 g) were homogenized in 10 mL Tris/HCl 0.1 M, 1 mM EDTA; pH 7.6 buffer and centrifuged 15 min at 10,000× *g* to isolate the supernatants for enzyme activity determinations. G6P-DH activity was determined by measuring the increase in absorbance by photometry (340 nm) [15].

### 2.10. Islet Isolation

After euthanasia, pancreatic islets were obtained from the whole pancreas by collagenase digestion as previously described by our group [17,18].

### 2.11. Insulin Secretion

Islets isolated from each experimental condition (run in triplicated) were incubated for 60 min at 37 °C in Krebs–Ringer bicarbonate buffer (KRB), 1% (*w*/*v*) BSA, pH 7.4 in the presence of different glucose concentrations (3.3, 8.3 or 16.7 mM). Thereafter, aliquots of the incubation medium were collected for insulin quantitation by radioimmunoassay as described in Section 2.3.

### 2.12. Total RNA

Islet and liver total RNAs were isolated employing TRIzol Reagent (Gibco-BRL, Rockville, MD, USA) [30]. Integrity and quality of the obtained RNA were evaluated by agarose-formaldehyde gel electrophoresis and the 260/280 nm absorbance ratio. Possible DNA contamination was discarded by treating samples with DNase I. SuperScript III and 1 μg of total RNA as template were used for performing reverse transcription-PCR.

### 2.13. Analysis of Gene Expression by Real-Time Polymerase Chain Reaction (qPCR)

For cDNA amplification, 10 ng of each cDNA sample were mixed with FastStart SYBR Green Master mix in the iCycler 5 (BioRad) and run in 40 cycles of 95 °C for 30 s (denaturation), 62 °C for 30 s (annealing) and 72 °C for 45 s (extension). Sequences of oligonucleotide primers used in the study are described in Table 1. All amplicons were designed in a size range of 90–250 bp. Melting curve analysis was employed for checking the reaction specificity. Results are shown as relative gene expressions normalized to the β-actin housekeeping gene. For that purpose, Qgene96 and LineRegPCR software (University of Basel, Basel, Switzerland) were used [31].

### 2.14. Statistical Analysis

ANOVA and Dunnett’s post-test for multiple comparisons and Bartlett’s test to assess variance homogeneity were used for statistical analysis. Data are expressed as means ± SEM and the differences between experimental groups were considered significant when *p <* 0.05.

## 3. Results

### 3.1. Body Weight and Water Intake

As can be seen in Table 2, weekly body weight increment was similar in the experimental groups (F and C) at all-time points (1, 2, and 3 weeks). Caloric intake was also comparable between C and F animals every week (Table 2); however, F rats consumed larger volumes of drink and less solid food than C rats, resulting in different percentages of daily nutrient intake in the groups (for C, F1, F2 and F3, the percentages of individual nutrient consumption were carbohydrates: 45%, 61%, 62% and 60%; proteins: 43%, 31%, 30% and 31% and lipids: 12%, 8%, 9% and 9% respectively).

### 3.2. Serum Glucose, Insulin, and Triglyceride Levels

Serum glucose levels at any time of treatment did not attain significant differences (Table 3). No differences were detected in insulin levels in F1 and F2 vs. C1 and C2 rats respectively (Table 3); however, they were significantly higher in F3 vs. C3 rats (*p* < 0.05). Consequently, whereas HOMA-IR and HOMA-β indexes were similar in C1 vs. F1 and C2 vs F2 rats, they were higher in F3 compared to C3 rats (*p* < 0.001), demonstrating that these rats are in an insulin resistant state. Similarly, the decrease in hepatic insulin sensitivity in F rats attained statistical significance only at the third week of treatment (Table 3). However, triglyceride levels were significantly higher in F compared to C animals at all periods of treatment evaluated (Table 3).

### 3.3. Liver Protein Carbonyl Groups and Reduced Glutathione (GSH)

F2 and F3 rats evidenced significantly higher protein carbonyl content than C2 and C3 animals (Figure 1A). Conversely, GSH content showed a significant reduction in F2 and F3 rats compared to their controls (C2 and C3) (Figure 1B). No differences were found in F1 vs C1 animals.

### 3.4. Liver Gene Expression (by qPCR)

Figure 2 shows that SREBP-1c gene expression significantly increased in F2 and F3 rats compared to C2 and C3 respectively. However, its target genes, FAS and GPAT, showed higher expression only in F3 vs, C3 animals. Furthermore, although G6Pase and G6P-DH gene expressions were higher in F rats at any time point, this difference only reached statistical significance for G6Pase in F3 rats and for G6P-DH in F2 animals compared to their controls.

### 3.5. Content of Liver Triglyceride and Glycogen

Hepatic triglyceride content was higher in F rats at all treatment times (Figure 3A). However, liver glycogen deposit in F rats increased only after the second week of treatment (Figure 3B).

### 3.6. Liver Glucokinase, G6P-DH, and G6Pase Activities

While glucokinase and G6P-DH activities were increased in F animals at any time of treatment and showed a similar pattern (Figure 4A,C), G6Pase evidenced higher activity in F2 and F3 animals compared to C rats (Figure 4B).

### 3.7. Glucose Stimulated Insulin Secretion

Islets isolated from C and F animals treated for 1, 2, or 3 weeks, when challenged with increasing amount of glucose (from 3.3 mM to 16.7 mM final concentration), secreted insulin in vitro in a dose-dependent way (Figure 5A–C). Although in basal conditions (3.3 mM glucose) no differences were found between F and C rats at any time point (1, 2, and 3 weeks), islets isolated from F3 rats with the highest glucose concentrations released significantly larger amounts of insulin than those isolated from C3 rats (Figure 5C).

### 3.8. Inflammation Marker Gene Expression

We found a significant increase in liver PAI-I, IL-1β and TNF-α mRNA levels in F3 compared to C3 animals (Figure 6A). Similarly, islet PAI-I and IL-1β gene expression became significantly higher in F3 compared to C3 rats (Figure 6B).

## 4. Discussion

It has been postulated that impairment in glucose metabolism in the liver is one of the earliest reactions to increased flow and availability of fructose [5,32,33,34]. We previously demonstrated that three weeks-FRD fed rats developed changes in liver glucose and lipid metabolism paralleling endocrine dysfunction (hyperinsulinemia, hyperleptinemia, higher plasminogen activator inhibitor-1 and lower adiponectin levels) and an insulin resistant state [14,15,16,17,18,19,20]. These rats also show increased oxidative stress markers in the liver and pancreatic islets [14,15,20]. Therefore, it is suggested that the gluco-oxidative stress could play a key role in the detrimental effects of FRD [16,20,35,36,37,38,39,40]. In this case, these alterations could be prevented by co-administration of antioxidant agents [35,36,37,38,39,40].

The current results demonstrated that these fructose-induced changes are not synchronic, the increase of serum triglyceride level being one of the first parameters to evidence the dysfunctions induced by the diet. This change recorded at 1 week of treatment was accompanied by a significant accumulation of triglycerides in the liver, thereby suggesting that lipid dysmetabolism is present at the beginning of the pathogenic sequence leading to long-term effects of fructose. Interestingly the lipogenic master gene regulator, SREBP-1c, increases in the liver at the second week whereas its target genes, FAS and GPAT, increased in the third week. FAS and GPAT are key enzymes involved in the synthesis of fatty acids (FAS) and triglycerides (GPAT) in the liver, thus in consequence in liver lipid deposit. After fructose uptake, the sugar is phosphorylated by fructokinase (mainly located in the liver), and thus initiating hepatic fructolysis. In the next step, fructose-1-phosphate is cleaved by aldolase B and triokinase activities, thus rendering triose phosphate intermediaries. These metabolites can enter directly into the lipogenic pathway as well as can act as activators of SREBP1c which in turn regulates hepatic gluconeogenesis and de novo lipogenesis gene expressions [33]. However, our data suggest that the increased intrahepatic triglyceride content observed after 1 week of treatment could be due mainly to non-hepatic lipid supply rather than intrahepatic synthesis. Complementarily, isotope tracer experiments have demonstrated that since 31–59% of ingested fructose is oxidized within 3–6 h after ingestion, the increment in serum triglycerides probably does not involve de novo lipogenesis [41]. An alternative mechanism described for enhanced circulating lipids after an acute fructose diet is a decrease in lipid uptake from peripheral organ [42].

The first carbohydrate metabolism related change observed in F rats was the increase in glucokinase activity in F1 rats. This fact is in line with the demonstrated rapid fructose conversion to glucose in the liver after acute oral fructose load [41]. This alteration was simultaneous with a significant increase in G6P-DH activity. As previously described, we hypothesized that this increase was a metabolic adaptation to fructose overload [14]. A similar increase in glucokinase activity was previously reported in dogs and humans challenged with fructose [43,44]. In this regard, it has also been described that dietary fructose triggers a fast increase in liver glucose uptake that depends on glucokinase activation [45]. Therefore, enhanced entry of carbons into glycolysis occurs: the earliest events after a fructose overload. Enhanced generation of G6P (the product of glucokinase activity) may in turn lend to an increase of G6P-DH activity, the other very early marker of carbohydrates dysfunction. Since G6P represents a branching point of glucose metabolic pathways, the initial overload could be redirected to the pentose phosphate pathway thereby providing a shortcut to detour carbons out of mitochondria to avoid respiratory chain overload and the consequent production of reactive oxygen species [14,15]. Oxidative stress markers do not increase with 1-week treatment. Furthermore, pentose shunt also provides electrons to reduce glutathione, thereby generating GSH, a potent cellular antioxidant. However, as long as the carbon supply continues, this first barrier against oxidative stress is overwhelmed, and the liver starts to accumulate oxidative stress markers, e.g., increased carbonyl groups in protein and decrease of GSH as observed in our F2 rats. Once this point is reached, two new mechanisms appear to redirect carbon glycolysis downstream: enhanced glycogen synthesis and increased G6Pase activity. In our model, both these changes were observed by 2 weeks of treatment. Regarding this point, the simultaneous increase in both glucokinase and G6Pase potentiated the entrance of glucose into an apparently futile cycle. However, glucose cycling has been described as a protective mechanism in islets of ob/ob mice [46] and animal models of hyperglycemia [47]. In our model, increased glucokinase activity evidenced at the first week of treatment triggered a substrate circle on this level, this mechanism apparently maintaining normoglycemia efficiently, at least in our experimental conditions.

Administration of FRD for 3 weeks induced peripheral IR evidenced by hyperinsulinemia and increased HOMA-IR and HOMA-β indexes. In a previous report, we demonstrated that FRD also triggers a decrease in islet gene expression of insulin receptor and its intracellular mediators’ [20], evidencing the establishment of an islet IR state. Our current results also showed a reduction in liver insulin sensitivity at the third week of treatment. Consequently, this decreased insulin sensitivity triggered a compensatory increase in insulin secretion. Accordingly, pancreatic islets isolated from fructose fed rats showed increased glucose-stimulated insulin secretion after only three weeks of fructose treatment.

We have previously reported that enhanced β-cell secretory response to glucose recorded in islets from F animals is associated with a loss of β-cell mass, mainly attributed to an enhanced apoptotic rate [16,18,19,20]. The molecular mechanism responsible for these changes includes an increase in both, endoplasmic reticulum and oxidative stress, together with mitochondrial dysfunction, and an enhanced inflammatory and glyco-lipotoxic state [48,49]. Concordantly, in our study, gene expression of TNF-α, PAI-1 and IL-1β was higher in F3 animals in both liver and pancreatic islets, suggesting that inflammation was already increased. IL-1β is a prominent pro-inflammatory mediator [50] that is also involved in the pathogenesis of type 2 diabetes (T2D) through activation of the NLRP3 inflammasome [51]. The enhancement in serum IL-1β level, recorded in diabetes has been suggested to contribute to the progression of inflammation and IR [52]. In this context, β cells chronically exposed to high glucose concentrations stimulates the production/secretion of IL-1β which can exert an autocrine effect on the survival and function of β cells [53,54] mediating harmful effects of glucose by inducing apoptosis [55]. The greater inflammatory process detected at 3 weeks of treatment strongly suggests that it could be involved as a key player in the detrimental effects of a fructose overload in the reduction of β-cell mass [16,18,19,20].

## 5. Conclusions

In conclusion, our results showed that changes triggered by fructose overload are markedly asynchronous. Whereas liver responded to the fructose overload as soon as one week of treatment, modifying its carbohydrate metabolism mainly through the glucose sensor glucokinase, endocrine pancreas takes longer to modify its function (3 weeks), which is determinant for the establishment of systemic IR. Since circulating triglycerides constitute the earliest distorted circulating marker and although liver fat deposit is evident with 1 week of fructose overload, it is possible that adipose tissue could be the first organ to be altered by a fructose diet. Since the establishment of an inflammatory state together with the enhanced oxidative stress and all the endocrine-metabolic dysfunctions found in our murine model of fructose overload are comparable to those reported in human pre-diabetes, detailed knowledge of the time-course of their appearance could help to propose new strategies to prevent its progression to T2D.

## Figures and Tables

**Figure 1 medicina-58-00008-f001:**
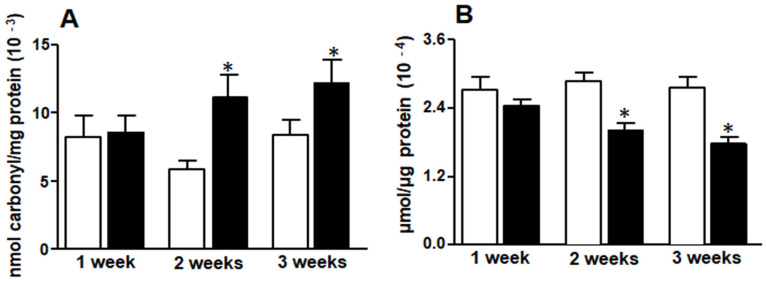
Liver protein carbonyl groups (**A**) and reduced glutathione (GSH) (**B**). White bars: control group (C rats), black bars: rats fed fructose-rich diet (FRD) (F rats). Values are expressed as means ± SEM (*n* = 12 rats per group) * *p* < 0.05 compared to each control at each time point.

**Figure 2 medicina-58-00008-f002:**
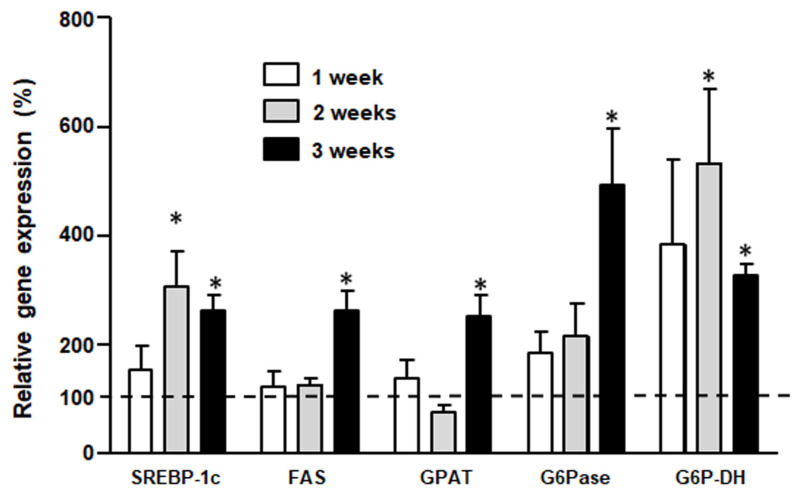
Liver gene expression, lipid and carbohydrate metabolic pathways. Bar plot represents percentage expression of each gene found in F rats compared to the expression (100%) of their respective controls (C rats) (dashed line). Values are expressed as means ± SEM (*n* = 12 rats per group) * *p* < 0.05 compared to each control at each time point.

**Figure 3 medicina-58-00008-f003:**
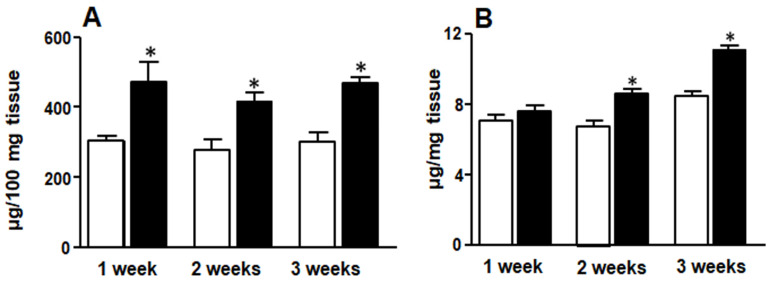
Liver triglyceride (**A**) and glycogen content (**B**). White bars: control group (C rats), black bars: rats fed FRD (F rats). Values are expressed as means ± SEM (*n* = 12 rats per group) * *p* < 0.05 compared to each control at each time point.

**Figure 4 medicina-58-00008-f004:**
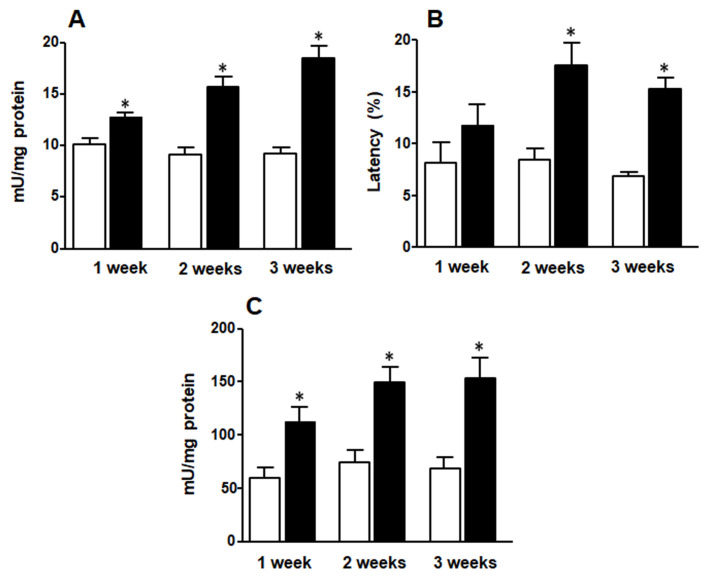
Liver glucokinase (**A**), G6P-DH (**B**) and G6Pase (**C**) activities. White bars: control group (C rats), black bars: rats fed FRD (F rats). Values are expressed as means ± SEM (*n* = 12 rats per group) * *p* < 0.05 compared to each control at each time point.

**Figure 5 medicina-58-00008-f005:**
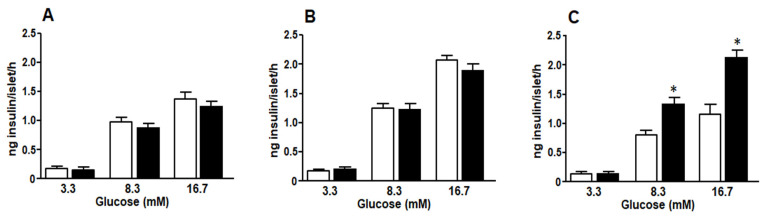
Glucose stimulated insulin secretion from pancreatic islets isolated from normal rats fed a standard control (white bars) or a fructose rich diet (black bars) for one (**A**) two (**B**) or three (**C**) weeks. Values are expressed as means ± SEM (*n* = 12 rats per group) * *p* < 0.05 compared to each control at each time point.

**Figure 6 medicina-58-00008-f006:**
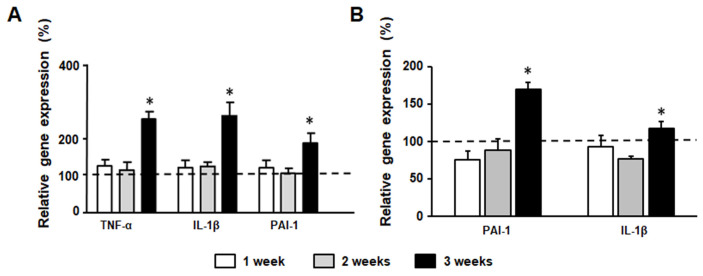
Inflammation marker gene expression. Liver (**A**) and islet (**B**) gene expression. Bar plots represent percentage expression of each gene registered in F rats compared to expression (100%) in their respective controls (C rats) (dashed line). Values are expressed as means ± SEM (*n* = 12 rats per group) * *p* < 0.05 compared to each control at each time point.

**Table 1 medicina-58-00008-t001:** Primer sequences.

Gene	GeneBank Accession Number	Sequences
β-actin	NM_031144.2	FW 5′-AGAGGGAAATCGTGCGTGAC-3′ RV 5′-CGATAGTGATGACCTGACCGT-3′
SREBP-1c	XM_213329.5	FW 5′-TTTCTTCGTGGATGGGGACT-3′ RV 5′-CTGTAGATATCCAAGAGCATC-3′
FAS	NM_017332.1	FW 5′-GTCTGCAGCTACCCACCCGTG-3′ RV 5′-CTTCTCCAGGGTGGGGACCAG -3′
GPAT	NM_017274.1	FW 5′-GACGAAGCCTTCCGAAGGA-3′ RV 5′-GACTTGCTGGCGGTGAAGAG-3′
G6Pase	NM_013098.2	FW 5′-GATCGCTGACCTCAGGAACGC-3′ RV 5′-AGAGGCACGGAGCTGTTGCTG-3′
G6P-DH	NM_017006.2	FW 5′-TTCCGGGATGGCCTTCTAC-3′ RV 5′-TTTGCGGATGTCATCCACTGT-3′
TNF-α	NM_012675.3	FW 5′-GGCATGGATCTCAAAGACAACC-3′ RV 5′-CAAATCGGCTGACGGTGTG-3′
IL-1β	NM_031512.2	FW 5′-ACAAGGAGAGACAAGCAACGAC-3′ RV 5′-TCTTCTTTGGGTATTGTTTGGG-3′
PAI-1	NM_012620.1	FW 5′-CCACGGTGAAGCAGGTGGACT-3′ RV 5′-TGCTGGCCTCTAAGAAGGGG-3′

FW: Forward primer; RV: Reverse primer.

**Table 2 medicina-58-00008-t002:** Body weight, food and drink intake and calories consumed at each time point.

	C1	F1	C2	F2	C3	F3
**Body weight gain (g)**	33.6 ± 3.3	29.1 ± 2.9	61.0 ± 6.6	77.4 ± 3.3	95.5 ± 8.1	94.25 ± 10.9
**Food intake (g/rat/day)**	18.6 ± 0.4	12.4 ± 0.6 *	18.7 ± 0.9	14.2 ± 0.8 *	19.6 ± 0.6	15.3 ± 1.0 *
**Drink intake** **(ml/rat/day)**	22.0 ± 0.1	51.3 ± 7.9 *	23.6 ± 1.3	64.7 ± 9.3 *	25.3 ± 0.6	58.1 ± 0.2 *
**Calories** **Kcal/day**	53.8 ± 3.9	56.6 ± 4.1	57.5 ± 2.5	63.5 ± 5.5	60.1 ± 5.9	65.1 ± 6.1

Values are expressed as means ± SEM (*n* = 12 rats per group) * *p* < 0.05 compared with each control at each time point. Initial and final body weight (g) for each group were: C1 185.3 ± 1.7 and 218.9 ± 2.2; F1 184.6 ± 1.1 and 213.7 ± 3.4; C2 196.3 ± 4.9 and 257.3 ± 10.6; F2 188.3 ± 2.2 and 265.7 ± 5.4; C3 188.3 ± 2.8 and 283.8 ± 4.0 and F3 188.0 ± 2.3 and 282.3 ± 8.0.

**Table 3 medicina-58-00008-t003:** Seric parameters and derived indexes at each time point.

	C1	F1	C2	F2	C3	F3
**Glycemia (mg/dL)**	115 ± 3	118 ± 4	103 ± 3	121 ± 6	117 ± 2	117 ± 3
**Insulinemia (ng/mL)**	0.5 ± 0.07	0.6 ± 0.06	0.6 ± 0.06	0.7 ± 0.06	0.6 ± 0.04	1.1 ± 0.20 *
**Triglyceridemia (mg/dL)**	103 ± 6	134 ± 11 *	112 ± 15	172 ± 15 *	102 ± 11	226 ± 31 *
**HIS**	0.29 ± 0.05	0.23 ± 0.02	0.26 ± 0.04	0.19 ± 0.02	0.23 ± 0.01	0.13 ± 0.02 *
**HOMA-IR**	3.5 ± 0.3	4.4 ± 0.5	3.8 ± 0.4	5.2 ± 0.5	4.3 ± 0.3	7.9 ± 0.6 *
**HOMA-** **β**	35.6 ± 3.5	41.9 ± 4.8	48.9 ± 5.0	48.7 ± 4.5	42.6 ± 4.0	81.4 ± 6.8 *

Values are expressed as means ± SEM (*n* = 12 rats per group) * *p* < 0.05 compared with each control at each time point (One, two or three weeks). Indexes formulas: Hepatic insulin sensitivity (HIS): k/[serum insulin (µU/mL)] × serum glucose (mg/dL), where k: 22.5 × 18); HOMA IR: serum insulin (µU/mL) × serum glucose (mM)/22.5); HOMA β: serum insulin (μU/mL) × 20/serum glucose (mM)] − 3.5).

## Data Availability

The data presented in this study are available upon request from the corresponding author.

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
