# Peer review of "Chronological Appearance of Endocrine and Metabolic Dysfunctions Induced by an Unhealthy Diet in Rats"

_medicina, 2021, doi:10.3390/medicina58010008_

Round 1

Reviewer 1 Report

Abstract

10% w/v

It misses G6P-DH and T2D in full

please correct « a-synchronic »

Introduction

It is missing FFA in full

Impaired glucose tolerance (IGT)

I would suggest « by oral fructose administration... »

2.3

I would suggest a simpler sentence like « serum triglyceride levels... »

Please correct to Mathews et al.

2.6

H2SO4    the positive charged ion comes at the first place

2.7

G6P-DH in full is not necessary since it was already defined

Please explain « five different experiments were done in triplicate ».

Please give a more clear explanation about the measurement of glucokinase

2.8

Please clarify 320 μL FeSO4  (diluted in H2SO40.15 N). It is not clear what the FeSO4 concentration is. It would better fit 320 μL of FeSO4 in H2SO40.15 N. The same for the sentence above were MoNH4 is also diluted in H2SO4.

it is missing a space between the number and the unit, e.g. 3 ml. Please correct for other situations along the text.

Results

3.3 « GSH content showed a reduction as for the second week. » Please make this this sentence more clear.

please uniform 3-week and three weeks as well for the other time points (weeks). It would improve the reading.

please uniform first week and F1. I would suggest use C1 and F1 instead of controls or rats from the first week. It makes the text more clear.

Discussion

Please add SREBP-1c in full when it appears on the text for the first time.

Please give a brief explanation about the role of FAS and GPAT genes.

Please add a very brief explanation about the effects of fructose  precursors on SREBP-1c and consequent induction of lipogenesis. I would suggest to avoid « nutritional regulators » to designate the precursors resulting from fructose metabolism.

there are comas and dots in red colour along the text.

References

It is missing the reference 30. Please correct the references after 30. Some reference numbers do not match with the scientific articles cited on the text.

Reviewer 2 Report

Summary

The authors aimed to evaluate the chronological sequence of events triggered by fructose administration to normal rats. Authors have found that fructose-rich diet induced a-synchronic changes involving early hypertriglyceridemia together with intrahepatic lipid deposits and metabolic disturbances from week one, followed by enhanced liver oxidative stress, liver and pancreas inflammation, pancreatic beta-cell dysfunction, and peripheral insulin-resistance registered at the third week.

Comments

  1. Daily consumed calorie data are shown in Table 2; however, the authors have not written in the methods section how it was calculated. Authors either should write the calculation method in the Methods part or show the related methods and data in the Supplementary material. The latter would have been better because there was no difference between the groups. In supplementary material, authors also can describe the composition of the chow. 
  2. Reduced solid food intake also means decreased vitamin and mineral intake compared to carbohydrate intake. Have the authors seen any symptoms of deficiency disease in the rats?
  3. Did the authors measure the daily amount of urine? The animals probably had polyuria. Is polydipsia due to increased urination or because the animals liked the sweet liquid?
  4. How was the 10% fructose solution prepared?
  5. Did the author measure the weight of the internal organs (e.g., liver) or different adipose tissues? Were visible signs of fatty degeneration on the surface of the liver?
  6. Authors should provide the starting and final body weight results, at least as Supplementary material.
  7. The authors have shown the liver glucokinase activity. Can they specify which enzyme (Hexokinase A, B, C (EC 2.7.1.1) or Glucokinase (EC 2.7.1.2)) is responsible for this activity? 
  8. Can the author provide information about the activity of other fructose metabolism-related pathways (e.g., enzymes like Fructokinase (EC 2.7.1.4) and Liver-type aldolase (EC 4.1.2.13) or Sorbitol dehydrogenase (EC 1.1.1.14) and Aldose reductase (EC 1.1.1.21))?

Round 2

Reviewer 2 Report

I accept the revised manuscript.